# Assessment of the Effectiveness of the Sonofeedback Method in the Treatment of Stress Urinary Incontinence in Women—Preliminary Report

**DOI:** 10.3390/jcm11030659

**Published:** 2022-01-27

**Authors:** Gabriela Kołodyńska, Maciej Zalewski, Anna Mucha, Waldemar Andrzejewski

**Affiliations:** 1Department of Physiotherapy, Wroclaw University of Health and Sport Sciences, 51-612 Wroclaw, Poland; waldemar.andrzejewski@awf.wroc.pl; 2Faculty of Medical Sciences and Health Sciences, University of Social and Medical Sciences in Warsaw, 04-367 Warszawa, Poland; 3Department of Gynaecology and Obstetrics, Faculty of Health Sciences, Medical University of Wrocław, 50-367 Wroclaw, Poland; zalewskim@interia.pl; 4Independent Public Health Care Center of the Ministry of the Interior and Administration in Wroclaw, Department of Gynaecology, 50-233 Wroclaw, Poland; 5Department of Genetics, Wrocław University of Environmental and Life Sciences, 50-375 Wroclaw, Poland; anna.mucha@upwr.edu.pl; 6Faculty of Health Sciences, University of Opole, 45-040 Opole, Poland

**Keywords:** electrostimulation, sonofeedback, stress urinary incontinence

## Abstract

Urinary incontinence is a common problem that affects postmenopausal women. This ailment has a negative impact on many aspects of life, significantly limiting everyday functioning related to professional work, physical activity or the intimate sphere. The aim of the study was to assess the effectiveness of the sonofeedback method in reducing the severity of urinary incontinence in postmenopausal women with a urinary incontinence problem. A total of 60 patients aged 45–65 with stress urinary incontinence, confirmed by a gynecologist, were qualified for the study. All persons qualified for the study were randomly assigned to study group A (*n* = 20), comparative B (*n* = 20) and control C (*n* = 20). Patients from group A were treated with sonofeedback of the pelvic floor muscles. In group B, the combined electrostimulation method was used with biofeedback training. Group C was a control group in which only the measured parameters were measured at the same time interval as those carried out in groups A and B. In all patients, three times: before the therapy, after the fifth procedure and after the end of the therapy, the incidence and severity of stress urinary incontinence were assessed—Gaudenz questionnaire and the intensity of urinary incontinence—a modified 1 h pad test. The obtained results have an application value giving the possibility of using effective therapy with sonofeedback in women in whom the recommended electrostimulation method cannot be used due to health and behavioral reasons.

## 1. Introduction

Stress Urinary Incontinence (SUI) is the most common type of incontinence and accounts for 50 to 88% of all types [1]. It is characterized by urine leakage caused by a sudden increase in abdominal pressure. Symptoms occur during sneezing, leaning forward, and heavy lifting [2]. It is caused by weak pelvic floor muscles and musculoskeletal system insufficiency [3]. The most common causes of SUI include childbirth, trauma, endocrine disruption or lowering of the reproductive organs and surgery in the lower abdominal region [4].

According to recent data, SUI is twice as likely to affect women [5]. In its report, ICS states that this problem affects between 5% and 69% of the population [6]. It is estimated that the actual prevalence of incontinence is much higher, but numerous patients do not report incontinence due to its embarrassing nature and the mistaken belief that it is a natural phenomenon, inextricably linked to the aging process [7,8].

Many classifications of incontinence risk factors are presented in the available literature. The most common of these include genetic, social, cultural, and ethnic causes due to differences in anatomy in women of different races [9]. Menopause is considered an important risk factor. The available studies suggest that there is then a sharp increase in diuretic symptoms, which may be due to estrogen deficiency in the blood. This results in the lowering of the urethra and its shortening, sometimes up to 1–2 cm. In addition, postmenopausal women often experience weakening of the muscles that form the bladder walls, as well as the muscles responsible for urine excretion, which further aggravates incontinence symptoms [10].

It is accepted that conservative management should play a major role in the first-line treatment of urinary incontinence. Surgical treatment should only be considered in cases of significant disease progression or in patients with pelvic floor stability abnormalities [11,12].

The latest physiotherapy methods offer many options to both reduce and even prevent the presence of adverse effects of urinary incontinence [13]. The primary physiotherapeutic method used in patients with urinary incontinence is pelvic floor muscle exercises [14,15]. Because of the difficulty in learning to exercise the pelvic floor muscles, biofeedback has been used for training. It involves providing the patient with feedback on changes in their physiological state. This makes it possible to learn to consciously modify functions that are not usually controlled, such as brain waves and muscle tension. One of the techniques using biological feedback to treat stress urinary incontinence is sonofeedback. There are studies in the literature supporting the effectiveness of ultrasound-guided treatment of stress urinary incontinence [16,17,18]. Using this method, it is possible to observe muscle activity in real-time with the help of an image seen on an ultrasound scanner. Using the ultrasound device, the physiotherapist teaches the patient to activate a specific muscle or an entire muscle group [19]. With this innovative method, pelvic floor muscle reinforcement is achieved in patients with urinary incontinence [20]. Furthermore, ultrasound allows the physical therapist to evaluate some of the functions in these muscles. During the examination on the monitor image, the therapist can evaluate the bladder neck, the urethra, the anorectal angle (ARA) and the pubic conjunctiva, which is a fixed bony point allowing measurement, all in the sagittal plane [21]. The methodology for performing ultrasound imaging is standardized and must be the same for all measurements [22].

The primary reason for addressing the topic of this study is the small number of scientific reports that address the evaluation of sonofeedback efficacy in the treatment of urinary incontinence. Electrostimulation with biofeedback is currently the most commonly used physiotherapeutic method [23]. This method is one of the most commonly used physiotherapeutic treatments for SUI. There are scientific reports that support the effectiveness of electrostimulation with biofeedback in the treatment of urinary incontinence [24]. For many patients, however, the form of therapy itself is unacceptable, so there is a need to find a method that is both effective and tolerable for patients. According to scientific reports, sonofeedback, which is a less invasive form of therapy, may prove to be an equally effective treatment option.

The aim of this study was to evaluate the effectiveness of sonofeedback in reducing the severity of urinary incontinence in postmenopausal women with stress urinary incontinence.

## 2. Material and Methods

The study included 60 female patients diagnosed with stress urinary incontinence. They were qualified for the study by a gynecologist based on precisely defined inclusion and exclusion criteria and an ultrasound examination performed. The inclusion and exclusion criteria for all patients participating in the experiment were the same.

### 2.1. Inclusion Criteria

-stage II stress urinary incontinence confirmed by ultrasound examination and an interview;-age 45–65;-postmenopausal age;-no health contraindications to the use of therapy;-a history of stress urinary incontinence of more than 5 years;-obtaining informed written consent from the patient;-ability to perform sonofeedback training correctly (after prior training by a physical therapist).

### 2.2. Exclusion Criteria

-an implanted pacemaker;-lower urinary tract infections;-decreased nerve excitability;-difficulty cooperating with the examined person.

The study was conducted at the Gynecology Outpatient Clinic at the Independent Public Health Care Facility of the Ministry of Internal Affairs and Administration in Wrocław, Poland, between February and July 2019. The project was approved by the Bioethics Committee of the Medical University of Wrocław with the number KB-806/2018.

A total of 75 patients were eligible for the project. Based on the criteria for the study, 60 patients took part in the measurements. The participants were randomly assigned to one of three comparison groups. Randomization was carried out using computer-generated random numbers. We used simple randomization. The participants were randomly assigned to groups in a 1:1:1 ratio. Detailed data on the characteristics of patients enrolled into group shown in Table 1. Figure 1 is shown the flow chart of the patients at each stage of the study.

All eligible subjects were randomly assigned to study group A (*n* = 20), comparison group B (*n* = 20), and control group C (*n* = 20). Patients in group A received sonofeedback therapy for pelvic floor muscles. The electrostimulation method combined with biofeedback training was used in group B. Group C was the control group, in which only the measurements of the assessed parameters were performed at the same time interval as in groups A and B.

The following tests were performed in all subjects on three occasions: before therapy, after the 5th treatment, and at the end of therapy:Assessment of the presence and severity of stress urinary incontinence—the Gaudenz questionnaire. This questionnaire allowed the assessment of stress urinary incontinence and its severity. It consists of 26 questions asking about the severity of symptoms; possible causes of the disease; situations in which it comes to letting go; gynecological and surgical interview; medications you are taking, including hormones; dealing with the issue of day and night pollakiuria and urgent urgency. Patients received a questionnaire validated in Polish, and it is a reliable source of information on problems related to the occurrence of SUI [25,26].Assessment of incontinence severity—modified 1 h pad test. The test followed the guidelines in the literature and was designed to assess the severity of SUI symptoms. The test was performed according to the guidelines in the literature. During the test, the test person drank 500 mL of water and waited for 30 min. Then she marched and climbed the stairs for 15 min. Subsequently, the patient performed a special exercise program: she sat down and got up 10 times, provoked a cough 10 times, and ran for 1 min in place, she picked up the water bottle lying on the floor 10 times, washed her hands under running water for 1 min. After the test was completed, the therapist weighed the sanitary pad, which was placed on the underwear before the test, and thus assessed the amount of lost urine [27,28].

Depending on the eligibility of the group, different therapeutic management was applied to the patients. The therapy program for group A patients included sonofeedback of the pelvic floor muscles. Each time, patients were asked to drink 500 mL of water 30 min prior to training so that training would be performed on a partially filled bladder and thus the bladder would be more visible on the screen. The correctness of the training was demonstrated by the displacement of the bladder base observed on the ultrasound monitor [29]. During training, the patient was in a supine position in a gynecological examination chair. The therapist placed the probe in the perineal area and vulval vestibule. This positioning of the probe allowed a panoramic image of the pelvis minor to be obtained. During training, the patient was asked to perform maximal contraction of the pelvic floor muscles. Together with the therapist, the patient observed displacement of the base of the bladder on the monitor image, which indicated that the pelvic floor muscle contraction was performed normally. The training was carried out on the USG of General Electric Voluson S10, country of device origin: South Korea.

According to the literature data, the following sonofeedback parameters were used:-training time—30 min;-probe type—linear;-number of trainings—10;-number of repetitions/series—10 repetitions/10 series;-contraction/relaxation time—5 s/10 s;-break time between sets—30 s;-frequency—3.5 MHz;-therapy in the 2D option [30].

The therapy program for patients in group B included electrostimulation of the pelvic floor muscles. The electrostimulation procedures were performed in a supine position on a couch, with the lower extremities placed on a set of wedges designed to provide comfort to the patient and lead to relaxation of the pelvic floor muscles. Before starting the procedure, the therapist placed a vaginal electrode with ultrasound gel in the patient’s vagina. They then dosed the current until the patient thought she felt a distinct muscle contraction. At the same time, it was said that the patient could not experience pain or discomfort from excessive muscle contraction during the procedure. After setting the appropriate intensity, a 30-min electrostimulation treatment was performed. After each electrostimulation treatment, patients underwent 5 min of biofeedback training. Both electrostimulation and biofeedback training were carried out using the MyoPlus4Pro 4-channel device, country of device origin: Great Britain. According to the literature data, the following electrostimulation parameters were used:-treatment time—30 min;-frequency—20 Hz;-pulse duration and pause time in a 1:1 ratio; -pulse time—1 ms; -max. current—up to 100 mA;-number of treatments—10 [31].

The EMG biofeedback training consisted in correctly identifying the pelvic floor muscles and learning their isolated contraction. The patient performed the tension and relaxation of the pelvic floor muscles with the help of visual feedback. During the training, she was in the same position as during the electrostimulation treatment because the biofeedback training took place immediately after it was performed. With the help of an endovaginal electrode collecting a signal from the muscles, the pelvic floor muscle tension was visualized on the screen of the device in real-time in the form of a displayed electromyogram. The educational game EMG biofeedback, dedicated to the therapy of stress urinary incontinence, was used for the training. The biofeedback training took place in such a way that the patient could observe the displacement of the index, reflecting the strength of the muscle tension during the contraction of the pelvic floor muscles. During properly generated muscle tension, the indicator moved up. In the absence of tension, it did not change its position, and when the contraction force decreased before the indicated time, it decreased. The contraction time was 5 s. Then there was a 10-s pause. The duration of the single training session was 5 min.

Treatments were performed daily—Monday through Friday, for 2 weeks, at a fixed time of day in the afternoon. Group C was the control group; no treatment programs were implemented in patients in this group, only follow-up measurements. 

Statistical analysis was performed using R Project software. Analysis of the results started by checking that the distributions of the variables evaluated were consistent with a normal distribution. Because there was no such compliance (Shapiro–Wilk test, *p*-value < 0.05), the statistical significance of differences between groups was verified using Kruskal–Wallis nonparametric analysis of variance. When statistical significance was obtained, a post hoc test (NIR) was performed, showing which means within a group or between groups differed in a statistically significant manner. The significance level for all statistical tests was *p* < 0.05. Statistically significant results are highlighted in red in the paper.

## 3. Results

At the end of therapy, a reduction in passed urine, as assessed by a modified 1 h pad test, was observed in comparison group B (electrostimulation group with biofeedback training). A decreasing trend was noted in study group A (sonofeedback group). The reduction in values in the group receiving electrostimulation treatments along with biofeedback training (group B) was statistically significant. There was a slight increase in the weight of the pad in the control group (group C) (Table 2).

Analysis of the individual responses collected from the Gaudenz questionnaire showed the changes that occurred in patients from all three groups: A, B, and C. Table 3, Table 4, Table 5 and Table 6 show the results for responses in which statistical significance appeared.

Analysis of responses to the question: “Do you involuntarily pass urine?” showed an improvement in this regard after five interventions, compared to the state before therapy, only in group B. No changes were observed in groups A and C (Table 3).

Responses to the question “How often does it happen?” were intended to determine how often the patients enrolled in the project experienced urinary loss events. The table also shows how the frequency of episodes changed with the progression of therapy. A statistically significant difference in the reduction of these incidents was observed in the sonofeedback training group. No statistically significant differences were observed in the other groups (Table 4). 

Results for answering the question: “How much urine do you pass?” is shown in Table 5. When evaluated after five interventions, a statistically significant difference was observed between the electrostimulation group and the control group. After therapy, a statistically significant change was seen in the sonofeedback training group.

The results related to incontinence symptoms during coughing and sneezing and laughing were associated with patients’ responses to the question: “In what situations do you involuntarily pass urine? Statistically significant differences after therapy were observed only in the sonofeedback group (group A). The number of patients who developed symptoms in the indicated situations decreased significantly (Table 6).

## 4. Discussion

The latest scientific reports indicate that conservative treatment has a positive outcome in up to 80% of patients with stage I and 50% of patients with stage II stress urinary incontinence. Completely successful treatment with conservative methods is not possible in patients with stage III stress urinary incontinence. Nevertheless, conservative treatment is followed by improvement in the form of reduction in the severity of symptoms [32]. Pelvic floor muscle training constitutes an important part of the treatment in the case of a patient with stage I or II stress urinary incontinence. It is important that the patient be trained on how to properly perform the assigned tasks before beginning the training. It is most difficult for patients to feel their pelvic floor muscles properly. The most common problem in therapy is activating the abdominal pressor instead of the pelvic floor muscles [33]. If performed correctly, pelvic floor muscle training could reduce the number of incidents of involuntary urination.

In our study, incidents of involuntary urine passing occurred in all subjects, as this was one of the eligibility criteria for the study. Only some of the patients indicated that incontinence occurs involuntarily in them. Others were aware of the moment when urination occurred. Some of them were able to indicate when their incontinence would appear in advance, which was due to their knowledge of the factors affecting the onset of symptoms. In the study group subjected to sonofeedback training, 55% of the women described that they involuntarily passed urine before therapy, and after the therapy, this was 50% of the women. Based on the results, electrostimulation proved to be a more effective form of therapy than sonofeedback in this aspect [34]. 

The frequency of involuntary urination alone also plays a large role in the comfort of life. An extremely important observation in our study was that these incidents in 55% of patients in group A repeatedly occurred before the start of therapy. However, after it was completed, as many as 60% of the women indicated that their involuntary urination occurred infrequently or occasionally. These differences are statistically significant and demonstrate the effectiveness of the sonofeedback method. The results obtained in the comparison group (group B) and the control group (group C) showed no significant changes.

The amount of urine passed is also important in the daily functioning of people with incontinence. Prior to therapy, patients in the study group (sonofeedback group) specified that they were experiencing profuse involuntary urination. The results indicated that after only five training sessions, a statistically significant reduction in urine passing was observed in this group. Although in the group of electrostimulation with biofeedback training, there was a reduction in the number of patients who indicated the presence of profuse enuresis, these changes were not statistically significant. No change was observed in the control group either. The results show that both sonofeedback and electrostimulation reduce the amount of passed urine. Sonofeedback was more effective, however, as there was a statistically significant reduction in passed urine.

The available literature contains many reports that describe symptoms typical of stress urinary incontinence [35,36,37]. The literature emphasizes that the following provoking situations have a major impact on involuntary urination in patients with SUI: physical activity, laughter, and coughing or sneezing. In a 1999 study by Kuh et al. of 1333 middle-aged women with urinary incontinence, up to 90% of patients reported increased incontinence symptoms during exercise [38]. This is also confirmed by the results obtained in our study. The results show that beneficial changes occurred only in the study group. Physical activity, walking, and going up and down the stairs were already not a significant problem for the subjects before the therapy in either group, so no change was observed after the experiment. This may indicate that the activity of walking and going up and down stairs did not have a significant effect on the incontinence problem for the women enrolled in the project.

Post-treatment results confirmed in some of the aspects assessed that sonofeedback training had a similar effect on the patients’ lives, compared to electrostimulation with biofeedback training. Sonofeedback was more effective in aspects such as reduction of involuntary urination and cessation of involuntary urination during laughing, coughing and sneezing, which are symptoms characteristic of stress urinary incontinence. In contrast, electrostimulation with biofeedback training was more effective in reducing the amount of passed urine, assessed by a modified 1 h pad test. Analysis of the results suggests that both methods are comparably effective in reducing symptoms associated with SUI. It is noteworthy that in the patients of the study group, in all aspects, the results showed a positive trend compared to the initial values. The efficacy of the sonofeedback method is also indicated by the results obtained in the modified 1 h pad test. Although there were no statistically significant changes after the end of therapy, there was a reduction in passed urine compared to baseline. 

The presented results indicate the effectiveness of sonofeedback in the treatment of stress urinary incontinence symptoms in postmenopausal women. It should be noted that there was an improvement obtained in terms of reducing the symptoms typical of stress urinary incontinence. The fact that these conditions reduce the quality of life of patients and their families and significantly limit their functioning in society demonstrates particular importance. 

It is necessary to search for effective treatment methods, as the number of women with the problem of incontinence is predicted to increase every year. The results obtained in this study indicate that sonofeedback may be one of them.

## 5. Limitations

Our paper has some limitations. Our findings may rely on an insufficient number of subjects, and, therefore, this research requires a follow-up. However, our limitation was obtaining a homogeneous group of women who met the study inclusion criteria. To our knowledge, this is the first paper to analyze the relationships between sonofeedback and electrostimulation in the treatment of stress urinary incontinence in postmenopausal women. Another limitation of the study is the lack of an even one-month follow-up.

## 6. Conclusions

Pelvic floor muscle sonofeedback training reduced the severity of some incontinence symptoms in the studied postmenopausal women.

Based on the obtained results, it can be suggested that the sonofeedback method is as effective in reducing urinary incontinence symptoms as the standard method of electrostimulation with biofeedback training, as evidenced in all assessed aspects by obtaining an improvement in relation to the initial values. 

The obtained results demonstrate application value, providing the possibility of using an effective sonofeedback therapy in women in whom the recommended electrostimulation method cannot be used due to health or behavioral reasons.

## Figures and Tables

**Figure 1 jcm-11-00659-f001:**
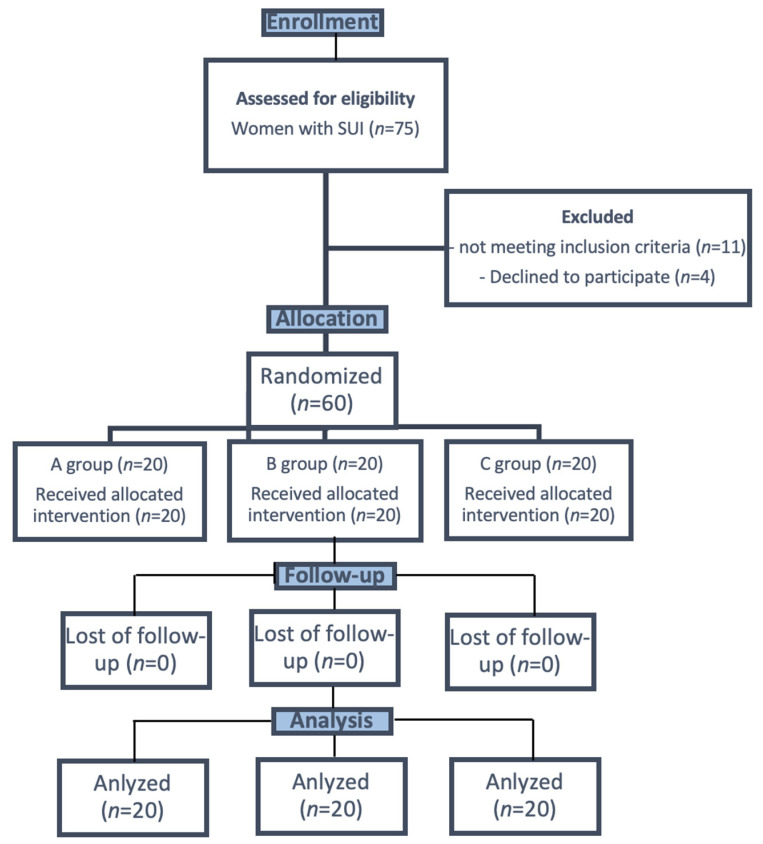
The flow chart of patients in the study. *n* = number of observations.

**Table 1 jcm-11-00659-t001:** Characteristics of patients enrolled in the study.

Number of People		*n* = 60
Age [years]	Min–max	45–65
	Mean	57.33
	Standard deviation	6.26
	Median	57.5
Body height [m]	Min–max	1.49–1.72
	Mean	1.61
	Standard deviation	0.05
	Median	1.62
Body weight [kg]	Min–max	48–100
	Mean	69.94
	Standard deviation	11.65
	Median	69.5
Body mass index [kg/m^2^]	Min–max	17.63–38.86
	Mean	27.07
	Standard deviation	4.69
	Median	26.74

**Table 2 jcm-11-00659-t002:** Comparison of changes in the amount of passed urine using a modified 1-hour pad test before, after the 5th treatment, and at the end of therapy.

	1-h Pad Test [g]			*p*-Value
	Before	After 5	After	Kruskal–Wallis Test
**Sonofeedback (A)**				
** *n* **	20	20	20	
**mean**	7.85	6.25	5.90	
**median**	6.00	6.00	6.00	
**minimum**	5.00	3.00	3.00	0.08851
**maximum**	24.00	10.00	9.00	
**SD**	4.56	1.68	1.12	
**v**	58	27	19	
**Electrostimulation (B)**				
** *n* **	20	20	20	
**mean**	16.60	11.30	7.85	
**median**	7.00	6.00	6.00	
**minimum**	5.00	4.00	3.00	0.03424
**maximum**	65.00	41.00	32.00	
**SD**	16.64	9.25	6.35	
**v**	100	82	81	
**Control (C)**				
** *n* **	20	20	20	
**mean**	6.70	6.50	7.15	
**median**	6.00	6.00	6.50	
**minimum**	3.00	3.00	3.00	0.3322
**maximum**	12.00	10.00	10.00	
**SD**	1.81	1.54	1.76	
**v**	27	24	25	
** *p* ** **-value**				
**Kruskal–Wallis test**	0.08688	0.1362	0.03013	

*n* = number of observations, SD = standard deviation, v = coefficient of variation.

**Table 3 jcm-11-00659-t003:** Presence of involuntary incontinence in all groups including progression of therapy.

	Sonofeedback (A)*n* = 20	Electrostimulation (B)*n* = 20	Control Group (C)*n* = 20	*p*-ValuePearson’s Chi-Squared Test
**Before therapy**	
**Yes**	11	15	12	


0.3934
**No**	9	5	8	
**After the 5th treatment**	
**Yes**	11	14	6	
				0.3800
**No**	9	6	14	
**After therapy**	
**Yes**	10	8	8	
				0.7622
**No**	10	12	12	
** *p* ** **-value**	
**Pearson’s Chi-squared test**	0.9352	0.04823	0.1495

**Table 4 jcm-11-00659-t004:** Incidence of involuntary incontinence in all groups including progression of therapy.

				*p*-Value
	Sonofeedback (A) *n* = 20	Electrostimulation (B) *n* = 20	Control Group (C) *n* = 20	Fisher’s Exact Test for Count Data
**Before therapy**				
**Every day**	4	4	5	
**Multiple times**	11	7	12	0.4822
**Practically constantly**	2	3	2	
**Rarely, occasionally**	3	6	1	
**After the 5th treatment**				
**Every day**	5	4	6	
**Multiple times**	4	4	6	
				0.8187
**Practically constantly**	2	2	0	
**Rarely, occasionally**	9	10	8	
**After therapy**				
**Every day**	6	5	7	
**Multiple times**	2	3	3	
				0.5508
**Practically constantly**	0	1	3	
**Rarely, occasionally**	12	11	7	
** *p* ** **-value**				
**Fisher’s Exact Test for Count Data**	0.01063	0.6318	0.1296	

**Table 5 jcm-11-00659-t005:** The amount of passed urine in all groups with respect to progression of therapy.

			*p*-Value
Sonofeedback(A)*n* = 20	Electrostimulation (B)*n* = 20	Control Group(C)*n* = 20	Fisher’s Exact Test for Count Data
**Before therapy**	
**A few drops**	7	6	5	
**Small portions**	12	7	13	0.1081
**Heavy wetting**	1	7	2	


**After the 5th treatment**	
**A few drops**	13	11	8	
**Small portions**	4	3	11	
				0.04732
**Heavy wetting**	3	5	1	
**After therapy**	
**A few drops**	9	11	8	
**Small portions**	11	7	10	
				0.5017
**Heavy wetting**	0	2	2	
***p*-value**	
**Fisher’s Exact Test for Count Data**	0.03106	0.1601	0.8020

**Table 6 jcm-11-00659-t006:** Presence of urine passing during laughter and coughing and sneezing in all groups.

					*p*-Value
		Sonofeedback (A) *n* = 20	Electrostimulation (B) *n* = 20	Control Group (C) *n* = 20	Pearson’s Chi-Squared Test
	Before therapy				
**Coughing and sneezing**	Yes	18	16	14	0.3457
	No	2	4	6	
**Laughter**	Yes	13	10	11	0.6218
	No	7	10	9	
	After the 5th treatment				
**Coughing** **and sneezing**	Yes	15	15	13	0.7201
	No	5	5	7	
**Laughter**	Yes	4	10	9	0.1270
	No	16	10	11	
	After therapy				
**Coughing** **and sneezing**	Yes	9	15	6	0.06495
	No	10	5	4	
**Laughter**	Yes	4	7	6	0.6752
	No	16	13	14	
** *p* ** **-value**					
**Pearson’s Chi-squared test**					
**Coughing and sneezing**		0.01583	0.9110	0.1193	

## Data Availability

The datasets used and/or analyzed during the current study are available from the corresponding author on reasonable request.

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
