# Peer review of "Assessment of the Effectiveness of the Sonofeedback Method in the Treatment of Stress Urinary Incontinence in Women—Preliminary Report"

_jcm, 2022, doi:10.3390/jcm11030659_

Round 1

Reviewer 1 Report

The authors presented interesting research aiming to extend conventional methods in the treatment of stress urinary incontinence in postmenopausal women. However, a number of queries arise from design and certain problems in methodology.

Comments:

Introduction

  1. Page 2. Paragraph 4. Lines 4-5. Statement that “This method is considered to be the most effective form of treatment for urinary incontinence” is under the question, since there is not enough evidence comparing electrical stimulation to other existing treatments such as drug therapy, pelvic floor muscle training plus vaginal cones, surgery, or different forms of electrical stimulation, to provide evidence-based guidance on which would be better, and for which women, in curing or improving SUI or in improving quality of life. Results of referred study/ies should be interpreted with caution as most of the comparisons were investigated in small, single trials. None of the published trials are large enough to provide reliable evidence.

Int Urogynecol J 2021, Int Urogynecol J. 2021 Aug 17.  doi: 10.1007/s00192-021-04928-2.

Material and methods

  1. Inclusion criteria: “Grade II SUI” needs explanation.
  2. What was the rationale for N of patients in the groups. Was the power analysis performed?  
  3. Gaudenz-Fragebogen questionnaire was originally developed in German and validated to support the differential diagnosis of female SUI and UUI. AA refer (25) to the Brazilian version of the G-F instrument. Has instrument been translated and adapted to the Polish culture assessed for reliability and validity be easy to understand and be applied? I.e., has translation, synthesis of translations, back translation, valuation of the synthetic version by a board of specialists and pre-test been performed before it was used in clinical study?  Questionnaire should be properly described in terms of full list of questions and scoring technique as English version was not referred.  
  1. Modification of 1-hour pad test needs explanation.
  2. It is mandatory to state what device brand and model was used in clinical trial and its origin as well. Here, ultrasound machine and electrostimulation unit.
  3. Details of 5 minutes biofeedback training after electrostimulation are missing.
  4. What statistical methodology/package was used?

Discussion

  1. AA in ref. 34 used ICIQ-UI, not G-F. Therefore, not eligible for comparison.
  2. G-F instrument do not assess quality of life but symptomatology, therefore AA should not conclude that, although globally comprehended, patients from the study have had impact on QoL

Limitations of the study

  1. No power analysis.
  2. Even 1 month follow-up is necessary for assessment of efficacy.
  3. No safety data.

References

  1. If not in language commonly used in journal, references should contain translation in brackets (Refs. 22, 26, 27).  

Author Response

Dear reviewer,

Thank you very much for your suggestions and comments.

Reviewer #1

Dear reviewer,

Thank you very much for your suggestions and comments.

Introduction

  • As suggested by the reviewer, we have changed the statement: "This method is considered to be the most effective form of treatment for urinary incontinence" to a more appropriate one:

"This method is one of the most commonly used physiotherapeutic treatments for SUI."

  • Thank you very much for pointing to the example of the article: "Efficacy and safety of electrical stimulation for stress urinary incontinence in women: a systematic review and meta-analysis"

Material and methods

  • We changed the statement "Grade II SUI" to "stage II stress incontinence confirmed by ultrasound examination and an interview"

- We wanted all groups to have an equal number of participants, hence n. Our study is a preliminary study because so far no publication has been published that objectively compares pelvic floor electrostimulation with pelvic floor muscle sonofeedback. The sample consisted of patients who reported to our Gynecological Clinic during the study, met the inclusion and exclusion criteria, and gave informed, written consent to participate in the project. We are aware that the groups are small and this is a limitation of our research.

  • Thank you for your comment on the evaluation methods used. We detailed the modified 1-hour pad test and Gaudenz questionnaire. In the conducted research, we used a questionnaire in the Polish language version, which is widely used in the national literature on the subject. We changed the position 25 of the literature in which we referred to the Polish-language article in which the Gaudenz questionnaire is described.

- We have added information about brands and models of ultrasound and electrostimulator.

- We have added a description of the biofeedback training that was carried out.

- Statistical analysis was performed using R Project software.

Discussion

We have removed the ref. 34, it was in fact incorrect to quote it at this point

We apologize for the overinterpretation of the G-F instrument. Of course, we know that it relates to symptomatology. We changed it at work

Limitations of the study

We added: Another limitation of the study is the lack of an even 1 month follow-up.

References

We added translation in brackets

Reviewer 2 Report

abstract:

-common problem that affects postmenopausal women: not only and why this choice?

-postmenopausal women : aged 45-65; aged 45 ? post menopausal...

introduction : 

post menopausal or perimenopausal ? you said: peri-menopausal women often

The primary reason for addressing the topic of this study is the small number of sci-entific reports that address the evaluation of sonofeedback efficacy in the treatment of urinary incontinence: could you describe sonofeedback exactly? 

methods section: sonofeedback how long, exercice: how many ; describe it like electrostimulation

Gaudenz questionnaire ? polish questionnaire from international questionnaire ? 

results :perhaps one table is possible with all this questionnaire results

discussion 

reduce +++ we know that pelvic floor muscle electrostimulation with muscle training is  effective and you give our result be already did it in result section so we need only discussion 

Author Response

Dear reviewer,

Thank you very much for your suggestions and comments.

Reviewer #2

Abstarct

- We decided on this group of patients because they constitute the largest group among our respondents.

The prevalence of SUI has been reported to increase at menopause and at menopausal age, and is more common in women than men, implicating menopause (Correia et al. 2009;Irwin et al. 2006;Iosif et al. 1981;Tinelli et al. 2010).

Correia S, Dinis P, Rolo F, Lunet N: Prevalence, treatment and known risk factors of urinary incontinence and overactive bladder in the non-institutionalized Portuguese population. Int Urogynecol J Pelvic Floor Dysfunct 2009, 20: 1481-1489. 10.1007/s00192-009-0975-x

Iosif S, Henriksson L, Ulmsten U: The frequency of disorders of the lower urinary tract, urinary incontinence in particular, as evaluated by a questionnaire survey in a gynecological health control population. Acta Obstet Gynecol Scand 1981, 60: 71-76.

Irwin DE, Milsom I, Hunskaar S, Reilly K, Kopp Z, Herschorn S, Coyne K, Kelleher C, Hampel C, Artibani W, Abrams P: Population-based survey of urinary incontinence, overactive bladder, and other lower urinary tract symptoms in five countries: results of the EPIC study. Eur Urol 2006, 50: 1306-1314. discussion 1314–5 10.1016/j.eururo.2006.09.019

Tinelli A, Malvasi A, Rahimi S, Negro R, Vergara D, Martignago R, Pellegrino M, Cavallotti C: Age-related pelvic floor modifications and prolapse risk factors in postmenopausal women. Menopause 2010, 17: 204-212. 10.1097/gme.0b013e3181b0c2ae

- When determining the qualifying age, we relied on the literature data and the age defined as the inclusion criterion by other researchers.In Poland the average age at menopause is 51 years. Most often, menopause occurs between the ages of 45 and 55 of a woman.

Introduction

- the sentence was supposed to apply to postmenopausal women. Thank you for paying attention. We have corrected the error

- we added detailed information about sonofeedback (similar to electrostimulation)

- we corrected the description of the Gaudenz questionnaire. Yes, it was an international questionnaire validated into Polish

Results

- when trying to create one table with the results for the Gaudenza questionnaire, it turned out to be illegible. The size and graphics of the table itself will change before the article is published.

Discussion

- we shortened the discussion as recommended by the reviewer

Round 2

Reviewer 1 Report

Revised manuscript meet the need in some items.

Minor: Country of devices' origin should be listed.

Major: Without follow-up assessment and comparison results are not reliable.

Major revision is mandatory.

Author Response

Dear reviewer, Thank you very much for your valuable comments and suggestions that significantly improved our manuscript. In the text of the manuscript, we added information about the country of origin of the devices used. Unfortunately, we do not have follow-up results at the moment. The manuscript we present is a Preliminary Report. We recognize that this is a limitation of our research. Thanks to the reviewer's opinion, we have added them in the Limitations section. Of course, in the future, we would like to carry out a follow-up assessment and conduct research on a larger group of patients. At the moment, our manuscript is the first to make this type of assessment, despite the fact that it concerns the treatment of a very important social problem. Therefore, we believe that the fact that we perform a triple assessment (before, after 5 treatments and after treatment) makes our research valuable and a valuable source of information for readers and other researchers. We have conducted our preliminary research and want to publish this article because we believe sonofeedback should be an alternative to pelvic floor electrostimulation. Our experience shows that there are many patients who are afraid of surgery with the use of electricity, especially in the intimate area. In addition, there are also many patients who cannot undergo electrostimulation surgery, which is why research on this topic is necessary, and unfortunately there is no such thing at the moment.

Reviewer 2 Report

you really improved your mansucript

Author Response

Dear reviewer, Thank you very much for your valuable comments and suggestions that significantly improved our manuscript.

Kind regards,
Authors
